# NME7 maintains primary cilium assembly, ciliary microtubule stability, and Hedgehog signaling

Menghui Ji*, Wenjuan Cui*, Qian Feng*, Jingjin Qi, Xinmin Wang, Hong Zhu, Wenqing Zhang, Wenxiang Fu ⓘ

**NME7 (nucleoside diphosphate kinase 7), a lesser studied member of the non-metastatic expressed (NME) family, has been reported as a potential subunit of the γ-tubulin ring complex (γTuRC). However, its role in the cilium assembly and function remains unclear. Our research demonstrated that NME7 is located at the centrosome, including at the spindle poles during metaphase and at the basal bodies during cilium assembly. Notably, a small fraction of NME7 localizes within the cilium. Detailed analysis of cilium assembly after NME7 knockdown and knockout revealed that NME7 is required for this process. NME7 knockout cells exhibited sensitivity to nocodazole, indicating its role in ciliary microtubule stability. In addition, NME7 deficiency impacted the Hedgehog signaling pathway, evident from reduced smoothened (Smo) fluorescence within primary cilia. This role of NME7 in Hedgehog signaling may depend on its nucleoside diphosphate kinase activity and γTuRC association. In conclusion, these findings enhance our understanding of the γTuRC roles in primary cilia in mammalian cells, highlighting the importance of NME7 in ciliary functions and signaling pathways.**

## Introduction

Centrosomes and cilia, which are essential elements in numerous cellular processes, have their formation and functionality deeply intertwined with the γ-tubulin ring complex (γTuRC). The function of the γTuRC in the pericentriolar material for the nucleation of microtubules is widely recognized. Recent structural studies have unveiled the structures of the γTuRC with unparalleled accuracy, shedding light on its role in microtubule nucleation (Aher et al, 2024; Brito et al, 2024; Dendooven et al, 2024). Moreover, recent discoveries have identified the presence of the γTuRC in other subpools, such as the centriolar lumen (Schweizer et al, 2021), outer centriole surface (Fu & Glover, 2012), and the subdistal appendages (Nguyen et al, 2020). However, our understanding of the role of the γTuRC in the cilium assembly and function is still limited. Data

showed that POC5 recruits augmin and the γTuRC to the centriole lumen for the centriole integrity and ciliogenesis (Schweizer et al, 2021). In another study, the percentage of ciliated cells increased with the knockout of Mzt2 and the N-terminal segment of GCP2. This was due to the γTuRC at the basal body recruiting microtubule-depolymerizing kinesin Kif2A to promote cilium disassembly (Shankar et al, 2022).

NME7 (non-metastatic expressed family member 7), belonging to the nucleoside diphosphate kinase (NDPK) family, was initially identified as a component of the γTuRC (Hutchins et al, 2010). NME7 interacts with the γTuRC through two NDK domains, and Arg-322 in the second NDK domain is crucial for binding (Liu et al, 2014). NME7 was also found to interact with CDK5RAP2, a human microcephaly protein that binds to the complex and participates in γ-tubulin centrosome attachment (Choi et al, 2010). NME7 was found to be required for centrosome-dependent microtubule nucleation, and this activity is dependent on its kinase activity in the γTuRC (Liu et al, 2014). However, it was later shown that NME7 had minor effects on MT nucleation activity of the γTuRC in vitro (Thawani et al, 2020). In a functional screen using RNA interference, NME7 emerged as a protein required for signaling within cilia and for the trafficking of proteins including smoothened to primary cilia (Lai et al, 2011). Recently, it has been observed that cells depleted of NME7 exhibited a slight accumulation in the G1 phase of the cell cycle, which was associated with an upsurge in the formation of cilia. Furthermore, these cells demonstrated elevated incidences of errors in chromosome segregation and an increase in micronuclei (Andersen et al, 2024).

Genetic studies have implicated the relationships between NME7 and ciliopathies. Nme7 knockout mice displayed situs inversus with left–right transposition of visceral organs and associated vasculature, in addition to hydrocephalus and excessive nasal exudates, which are signs of defective cilia (Vogel et al, 2010, 2012). Homozygous *Nme7* gene deletion in rats was semilethal and displayed multiple symptoms of primary ciliary dyskinesia, including hydrocephalus, situs inversus totalis, postnatal growth retardation, and sterility in both sexes (Sedova et al, 2021a). Genotyping and whole-exome analysis revealed a homozygous in-frame deletion of 34

Yunnan Key Laboratory of Cell Metabolism and Diseases, State Key Laboratory for Conservation and Utilization of Bio-Resources in Yunnan, Center for Life Sciences, School of Life Sciences, Yunnan University, Kunming, China

Correspondence: wenxiangfu@ynu.edu.cn
*Menghui Ji, Wenjuan Cui, and Qian Feng contributed equally to this work

amino acids in the second NDK domain existed in situs inversus totalis patients (Reish et al, 2016).

NME7 might also have other functions. A genome-wide association study implicated a significant association of NME7 with venous thromboembolism (Heit et al, 2012). Male rats with Nme7 heterozygous mutation showed decreased glucose tolerance, together with increased body weight, adiposity, and higher insulin levels (Sedova et al, 2021b). In the absence of the leukemia inhibitory factor, up-regulation of NME6 or NME7 could rescue the expression of stem cell markers and the formation of embryoid bodies (EB). This suggested that NME6/7 is crucial for the self-renewal of embryonic stem cells (Wang et al, 2012). Recently, it was shown that NME7 might activate WNT/$\beta$-catenin signaling to regulate one-carbon metabolism in hepatocellular carcinoma (Ren et al, 2022). However, whether these phenotypes are related to the centrosomal functions of NME7 is unclear.

In this study, we examined the localization of NME7 in detail and found that NME7 localizes to the centrosome and basal body. In addition, a small fraction of NME7 is present within the cilium. Despite the weak effects of NME7 knockdown on primary cilium assembly, we detected a significant reduction in the cilium assembly and reduced ciliary microtubule stability after NME7 knockout. Moreover, NME7 is required for Hedgehog signaling in cilia and this process likely depends on the NDPK activity and $\gamma$TuRC association of NME7. Together, our study has revealed the regulation and function of NME7 in primary cilia.

# Results

## NME7 localizes to the centrosome and basal body

During mitosis, the centrosome forms the two poles of the spindle. Previous studies have reported that NME7 is a centrosome protein. To verify this, we performed immunofluorescence experiments in RPE1 cells and detected the localization of NME7 during cell mitosis using the anti-NME7 antibody (Fig 1A). In addition to showing a diffuse distribution in the cytoplasm, NME7 has a strong staining at the centrosome. We further examined the localization of NME7 co-stained with $\gamma$-tubulin in both interphase and mitosis (Fig 1B). As shown, NME7 consistently co-localizes with the centrosomal protein $\gamma$-tubulin under different centrosome duplication and separation stages. With serum deprivation, RPE1 exits from the cell cycle and assembles primary cilia. ARL13b (ADP-ribosylation factor–like protein 13b) labels the ciliary membrane, and GT335 (polyglutamylated tubulin) indicates the ciliary axoneme. We examined NME7 with indicated centrosomal and ciliary markers in both non-ciliated and ciliated cells and confirmed that NME7 is localized at the basal body (Fig 1C and D). Together, these support that NME7 localizes to the centrosome and basal body.

Next, we tested the localization of exogenously expressed NME7. We then introduced a mouse or human NME7 overexpression vector into RPE1 cells, using a fused Flag tag at either the N or C terminus. Hereafter, we use Nme7 to refer to the mouse protein and NME7 for the human protein. Immunoblotting confirmed the overexpression of mouse Nme7 (Fig S1A) and human NME7 (Fig S1B). Using the CP110

antibody to specifically label centrioles, immunofluorescence analysis found that exogenous Nme7 is also located at the centrosome in normally growing interphase cells (Fig 1E). We confirm that exogenous Nme7 is located at the basal body during cilium assembly in serum-starved RPE1 cells (Fig 1F).

## A small fraction of NME7 localizes within the primary cilium

Although we noticed faint NME7 staining along the spindle microtubules besides the spindle poles during mitosis (Fig 1A), we next asked whether NME7 is also present within the cilium. Using high-resolution microscopy, we detected the staining of NME7, but not CEP164 (centrosome protein 164, a typical distal appendage marker), within the primary cilium (Fig 2A, also shown in Fig 1C and D). The endogenous NME7 tends to have stronger staining signals at the proximal part of the primary cilium. We then tested the localization of overexpressed mouse Nme7 and human NME7 and also confirmed the co-localization of Flag-tagged Nme/NME7 with primary cilium markers (Fig 2B–E). The NME7 staining pattern in most cases is shorter than in the entire cilium. It is worth mentioning that NME7 signals at the centrosomes are much stronger than within the cilium, and we only observed ~10% cells showing indicated ciliary staining. However, because of the limitation in microscopy resolution and high background caused by endogenous antibody quality or overexpression, we suggest that the percentages of cells with ciliary localization of NME7 could be much underestimated. For example, staining of exogenously expressed NME7 with the cilium can only be observed when the cilium was clearly viewed as emanating from the cell surface because of the high intracellular background.

## Effects of NME7 knockdown on primary cilium assembly

To explore the possible function of NME7 in the primary cilium assembly, we introduced NME7 shRNA into RPE1 cells to test changes in the cilium assembly. Western blotting showed that the NME7 protein level was markedly down-regulated after NME7 RNAi (Fig S2A) and the NME7 levels in knockdown cells were ~26.0% of those in control cells (Fig S2B). The immunofluorescence images showed that NME7 staining was weakened during division after NME7 knockdown (Fig S2C).

We further examined various cilium markers in RPE1 cells with NME7 knockdown, and representative immunofluorescence images of primary cilia co-labeled with ARL13b and Ac-Tub are shown (Fig 3A). Quantification of different primary cilium markers, including ARL13b, Ac-Tub, intraflagellar transport protein 88 (IFT88), and INPP5E (inositol polyphosphate-5-phosphatase E), revealed that NME7 knockdown did not cause a significant change in the primary cilium assembly (Fig 3B–E). Because there was a residual amount of the NME7 protein existed after RNAi, NME7 knockout cells were subsequently constructed for further analysis.

## NME7 knockout inhibits mature primary cilium assembly

We used CRISPR/Cas9 technology to knock out the NME7 gene, targeting its exons to generate indels and translation frameshifts, thereby achieving the null expression of NME7 in RPE1 cells.

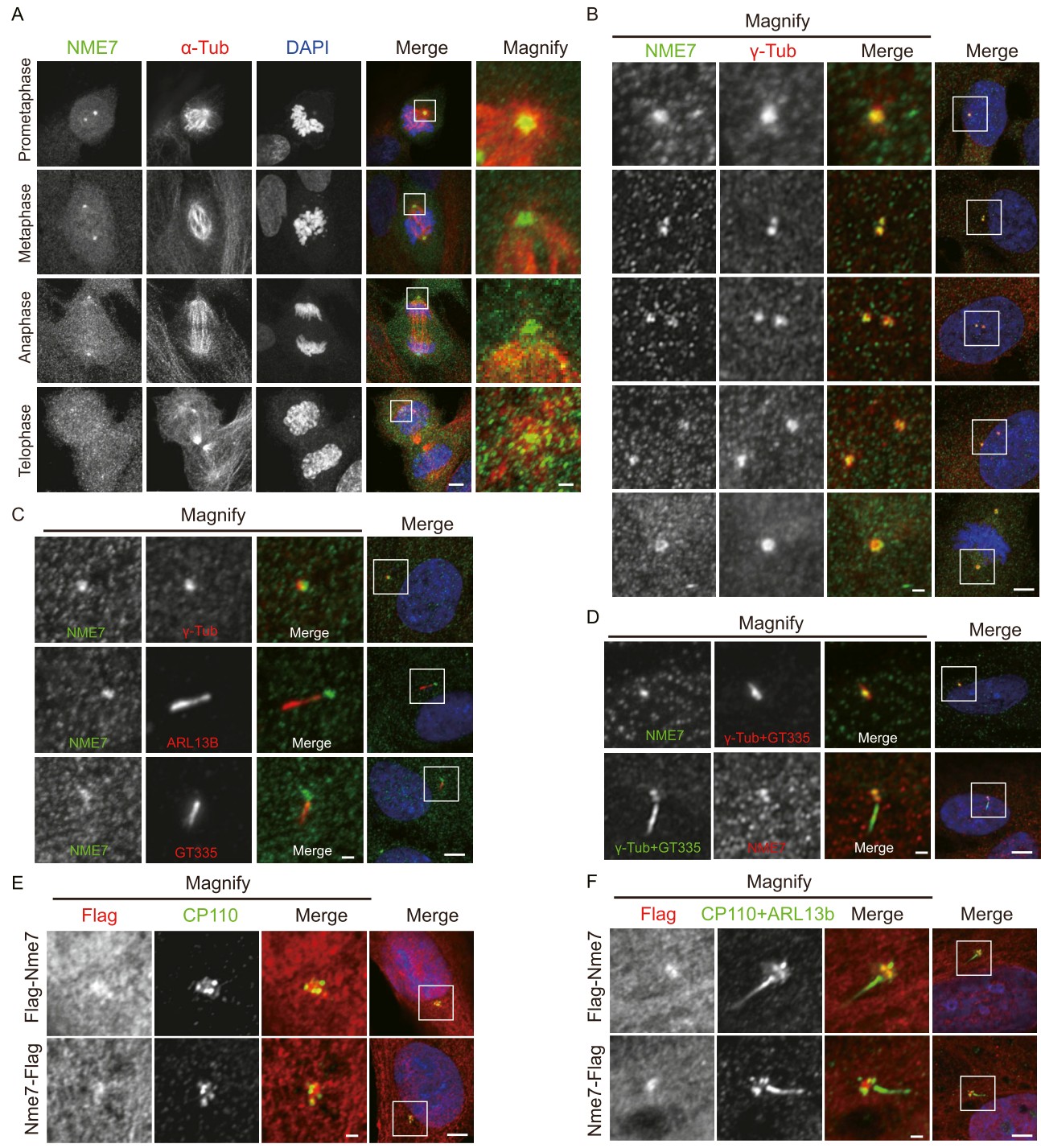

**Figure 1. NME7 localizes to centrosomes and basal bodies.**
**(A)** Endogenous NME7 localization during mitosis. Immunofluorescence images of RPE1 cells at prometaphase, metaphase, anaphase, and telophase are shown. NME7, green; α-tubulin (microtubule marker), red; DAPI (DNA), blue. **(B)** Co-staining of NME7 (green) with γ-tubulin (red) during different cell cycle stages. **(C, D)** Co-staining of endogenous NME7 with centrosome and cilium markers under serum-starved conditions. ARL13b indicates ciliary membrane, and GT335 (polyglutamylated tubulin) indicates axonemal microtubules. **(E, F)** Localization of exogenously expressed mouse Nme7 in normal-growing (E) and 48-h serum-starved (F) RPE1 cells. Flag-Nme7 and Nme7-Flag indicate the mouse Nme7 protein with a Flag tag at the N and C terminus, respectively; CP110, centriole marker. White boxes in (A, B, C, D, E, F) highlight magnified regions. Scale bars in unmagnified and magnified images are 5 and 1 μm, respectively.

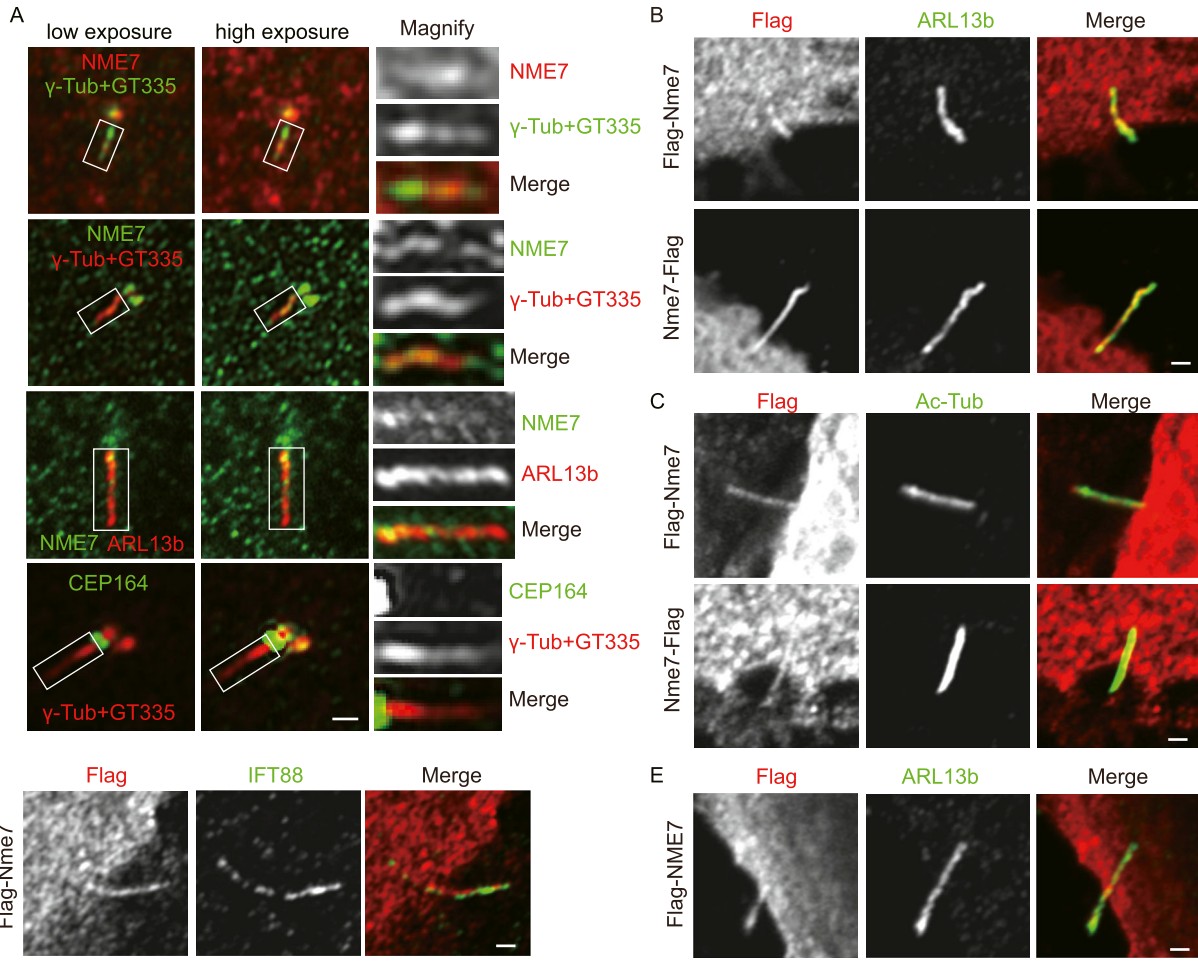

**Figure 2.  Small amount of the NME7 protein localizes to the primary cilium.**
**(A)** Staining of endogenous NME7 and other ciliary markers. Low- and high-exposure images show the localization of NME7 rather than CEP164 (distal appendage marker) along the cilium. White boxes highlight magnified regions. **(B, C, D)** Overexpressed mouse NME7 at the primary cilium. Flag indicates overexpressed mouse Nme7, red; ARL13b (B), Ac-Tub (C), and IFT88 (D) are primary ciliary markers in green. **(E)** Immunofluorescence image showing the localization of overexpressed human NME7 at the primary cilia. Flag-NME7, red; ARL13b, green. Scale bars in (A, B, C, D, E), 2 μm.

We designed two single-guide RNAs (sgRNAs): g1 and g2, and have obtained two single-clone cell lines. Sequencing confirmation of the genome DNA samples revealed an "A" base insertion in exons 3 (Fig S3A) and 5 (Fig S3B) in two clones, respectively. Western blotting also confirmed the ablation of NME7 expression in two NME7 KO clones (Fig S4A). In addition, NME7 localization in mitotic cells by immunofluorescence was diminished after NME7 knockout (Fig S4B). Together, these confirmed that two NME7 KO clones were successfully generated.

RPE1 cells begin to assemble primary cilia after serum starvation. To precisely test whether NME7 affects the occurrence of primary cilia, we analyzed the assembly of primary cilia both before and after serum starvation (Fig 4A and B). As expected, NME7 staining at the basal bodies was completely diminished after NME7 KO (Fig 4A). The morphologies of various markers for cilium (ARL13b, Ac-Tub, IFT88, and GT335) and centrosome (CEP164, centrosome protein 164) were not affected (Figs 4A and S4C–E). Quantification of the centriole distal appendage marker CEP164 also did not reveal any difference between control and NME7

knockout cells (Fig S4F). Quantification of GT335 and IFT88 revealed significant reductions in the cilium assembly upon NME7 knockout (Fig 4C and D). To maximally exclude the effects of the cell cycle on cilium assembly, cell density was maintained at 30–50% and fresh medium was changed 24 h before the SS 0-h assay point to enable active proliferation of the cells. Using ARL13b and Ac-Tub as markers, a time-course analysis of primary cilium assembly post-serum starvation also indicated reductions in the cilium assembly 48 h post-serum starvation (Fig 4E and F). However, no difference in the cilium assembly between control and NME7 KO cells was detected at 0–24 h post-serum starvation. Next, we performed rescue experiments using mouse Nme7 expression because the gRNA can recognize human NME7 cDNA and the mouse Nme7 protein is highly homologous to the human NME7. As expected, the reduction in the cilium assembly viewed by ARL13b in NME7 knockout cells was rescued by exogenously expressed Nme7 proteins (Fig 4G). Together, these data indicate that NME7 is required for proper cilium assembly at the mature stage.

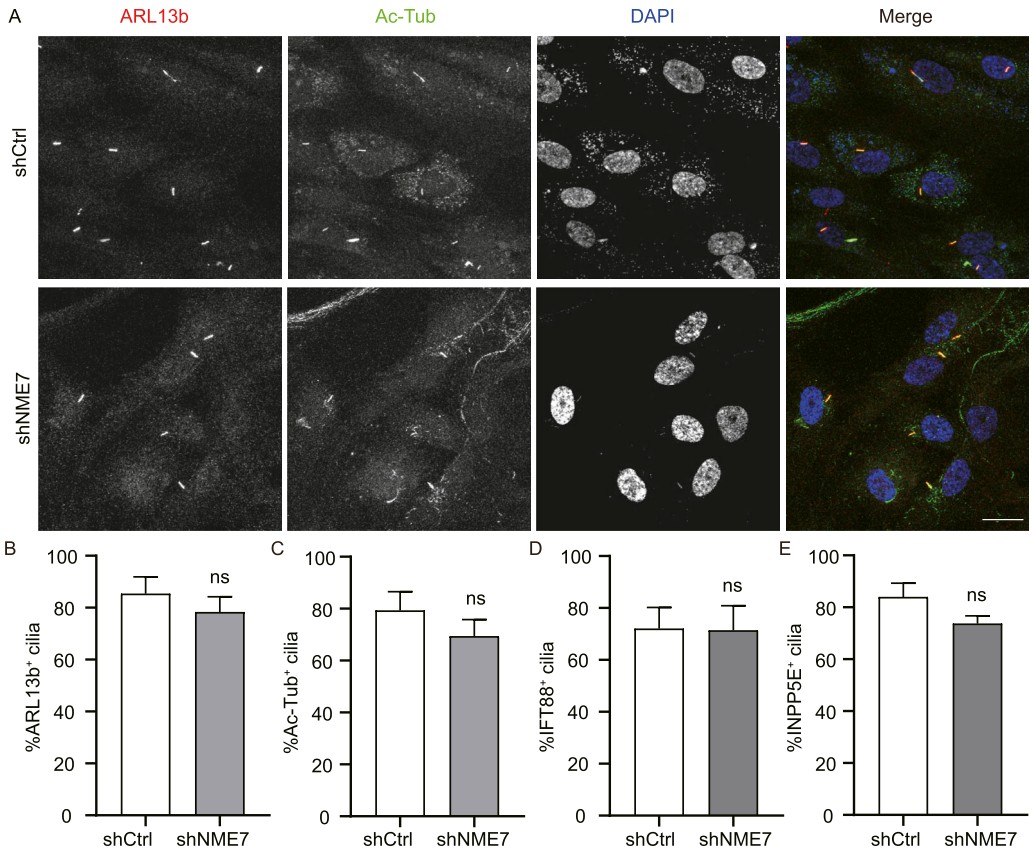

**Figure 3. Effects of NME7 shRNA knockdown on primary cilium assembly.**
**(A)** Representative immunofluorescence images of primary cilia in NME7 shRNA–interfered RPE1 cells after serum starvation for 48 h. ARL13b, red; Ac-Tub, green; DAPI, nucleus in blue. Scale bar, 10 μm. **(B, C, D, E)** Statistics of cilium occurrence in NME7 shRNA–interfered cells as in (A). Percentages of ARL13b$^+$ (B), Ac-Tub$^+$ (C), IFT88$^+$ (D), and INPP5E$^+$ (E) cilia are quantified.

## NME7 knockout reduces ciliary microtubule stability

Nocodazole can bind to β-tubulin and is a rapidly reversible microtubule polymerization inhibitor. Different concentrations of nocodazole treatment will induce different degrees of microtubule depolymerization, will block cells in the G2/M phase, and may lead to cell death. To verify the effects of NME7 on stability of mitotic spindle microtubules, we next tested the sensitivity changes of spindle microtubules in NME7 KO to the microtubule-depolymerizing drug nocodazole. Therefore, RPE1 cells and NME7 KO cells in the presence of different concentrations of nocodazole for 2 h were analyzed (Fig S5A and B). The spindle microtubules underwent different degrees of depolymerization with the treatment of 0–500 ng/ml nocodazole; however, the sensitivity of spindle microtubules to nocodazole in NME7 KO cells was similar to that in WT cells.

Axonemes are one of the most stable microtubule-based structures we know of, as axonemes are resistant to nocodazole and cold, which can induce complete depolymerization of cytoplasmic microtubules (Sharma et al, 2011). To test the axoneme stability, control and NME7 KO RPE1 cells were serum-starved for 48 h and then treated with 1 μg/ml of nocodazole to depolymerize microtubules for 1 h. Representative images of cilium staining and

EdU incorporation are shown (Fig 5A and B). Quantification of the ratio of cilium occurrence in nocodazole-treated versus untreated cells indicated that cilia in NME7 KO cells were more sensitive to nocodazole treatment compared with the control (Fig 5C). No significant change in cilium length was observed in the NME7 KO cells compared with the control (Fig 5D). Percentages of the S phase (indicated by EdU incorporation) and mitosis were not affected by NME7 knockout or nocodazole treatment in the cilium assembly assay conditions (Fig 5E and F). As a note, because the cells were serum-starved for a long time (48 h) to induce quiescence and only treated with nocodazole for a short period (1 h), the percentages of mitotic cells in all cases were low (<2%). Together, these results suggest that NME7 may maintain ciliary microtubule stability and this is unrelated to the cell cycle changes.

## NME7 regulates Hedgehog signaling

Given that primary cilia serve as the structures for transducing the Hh signaling pathway, we investigated whether NME7 directly influences Hh signaling. Both wild-type and NME7 KO RPE1 cells were subjected to serum starvation for 24 h, and the cells were further kept in the serum-starved medium without or with the Smo (smoothened) agonist SAG for an additional 24 h. As shown (Fig 6A),

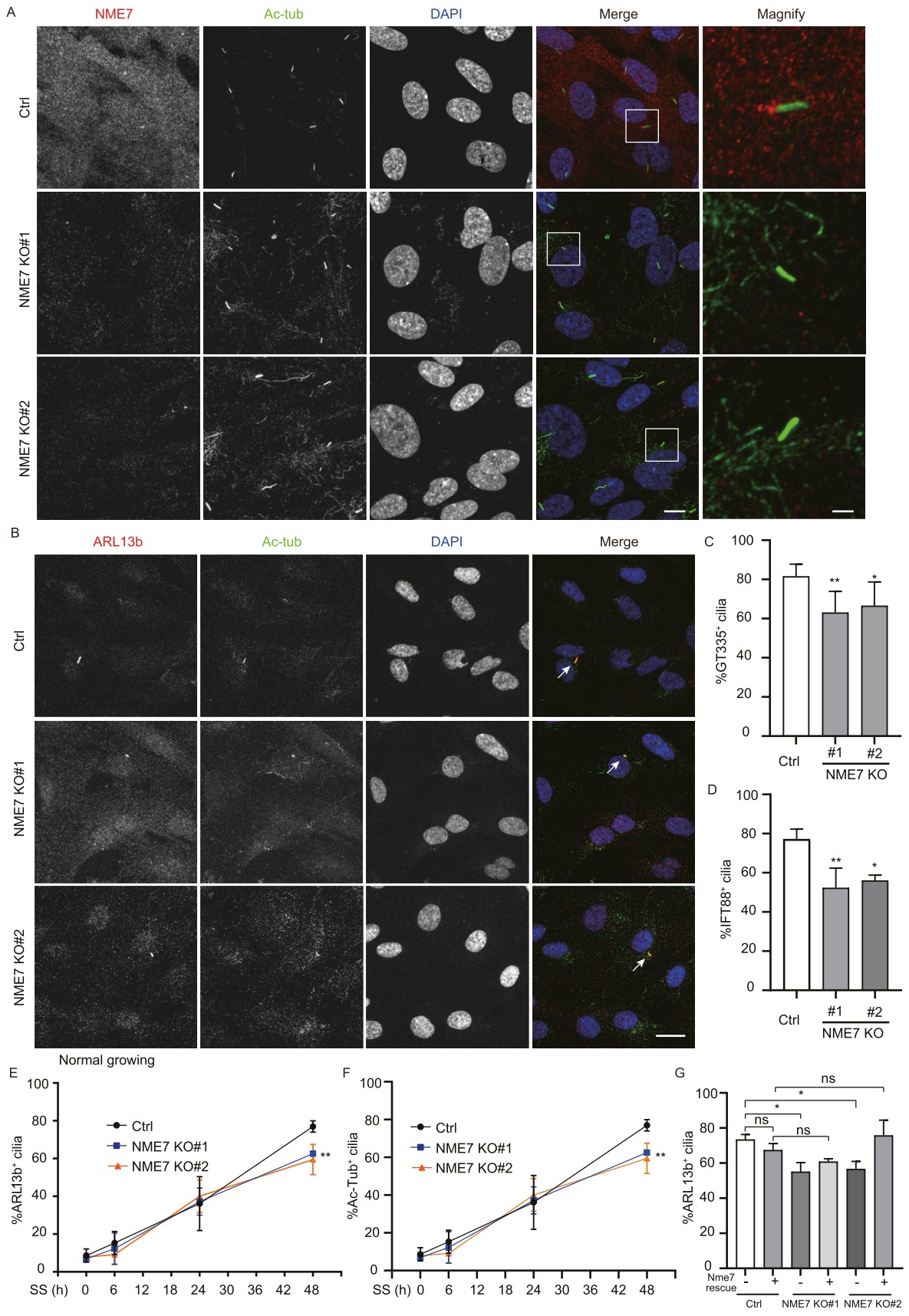

SAG treatment in control cells led to translocation of Smo into the cilium, indicative of Hedgehog signaling activation. However, this process was much dampened in NME7 KO cells. We quantified the percentages of cells exhibiting strong Smo fluorescence, and the data support that NME7 KO impedes the entry of Smo into the cilium upon SAG treatment (Fig 6B and C). To more precisely evaluate the roles of NME7 in Hedgehog signaling, we measured the Smo intensity in individual cells normalized against cilium length and the data also clearly showed that NME7 KO led to the reduction in Hedgehog signaling induction upon SAG treatment (Fig 6D).

To further explore the molecular function of NME7 in the primary cilium, we constructed two Nme7 mutants: H225F (homologous to H206 in human) and R341A (homologous to R322 in human). The human counterparts of H225F and R341A were shown to disrupt NDPK activity and γTuRC integrity, respectively (Liu et al, 2014). We then tested the functions of these two mutants and the WT in rescuing the Smo translocation phenotypes in two NME7 KO clones (Fig 6E and F). Data suggested that the WT could rescue the Hedgehog signaling defects caused by NME7 knockout. As shown in the figure, the two WT rescue groups were not significantly different from the control group. However, the rescue groups for both KO clones, whether with H225F or R341A mutant, were significantly lower than the control group. The quantified average value in the H225F-KO1 group appeared higher than in the KO1 group, likely because of experimental variations caused by the relatively low number of cells quantified in the H225F-KO1 group. Together, these results indicate that NME7 is important for ciliary Hedgehog signaling, likely requiring its NDPK activity and γTuRC formation capability.

# Discussion

In summary, our research has demonstrated that NME7 plays a role in the assembly and function of primary cilia. We discovered that NME7 is located at the centrosomes during various cell cycle phases, with most of NME7 found at the basal body when primary cilium assembly is induced by serum starvation. Interestingly, we also observed that a small proportion of both endogenous and exogenous NME7 is present within the primary cilia, although the functional implications of this require further investigation. Although the localization of the γTuRC at the cytoplasmic microtubules is well recognized, a recent cryo-EM study indicated that NME7 could be present in the doublet microtubules of the human respiratory motile cilia (Gui et al, 2022). We hypothesize that NME7 together with the γTuRC may also associate with axonemal microtubules in primary cilia, but this definitely requires more evidence in the future.

Although the knockdown of NME7 via shRNA had limited effects on primary cilium assembly, we proceeded to generate NME7 knockout RPE1 cells for a more detailed analysis. This revealed that NME7 knockout led to a reduction in cilium percentages at later stages of assembly. Although we did not observe significant effects of NME7 knockout on spindle assembly or spindle microtubule sensitivity to nocodazole treatment, we propose that the stability of the cilium axoneme is compromised after NME7 knockout. Lastly, we demonstrated that Hedgehog signaling was diminished after NME7 knockout, as evidenced by the reduced translocation of Smo upon SAG treatment.

In normal cells, the Hh pathway begins with the binding of the Hh ligand to the transmembrane receptor PTCH1, thereby alleviating its inhibitory effect on Smo. The activated Smo moves to the primary cilia, ultimately activating the GLI family member transcription factors, which then enter the cell nucleus to activate the expression of downstream target genes (Briscoe & Therond, 2013). However, in cells lacking NME7, the action of Smo is weakened and the occurrence rate of primary cilia decreases, preventing Smo from moving to the primary cilia and activating the expression of downstream transcription factors, leading to an interruption in the transduction of the Hh pathway signal. Importantly, our data suggest that the role of NME7 in Hedgehog signaling may require both NDPK activity and γTuRC association. Thus, it is reasonable to hypothesize that NME7 may locally produce GTP via γTuRC association to perform its functions. Although further research is needed to fully understand the function of NME7, this work helps us understand the regulation of cilium formation and function in mammalian cells.

Many tumors usually lack cilia, and primary cilia can either mediate or inhibit the formation of Hh pathway–dependent tumors depending on the initial carcinogenic event (Han et al, 2009; Wong et al, 2009). Although NME7 may be involved in hepatocellular carcinoma (Ren et al, 2022), whether the oncogenic roles of NME7 could be related to cilium and Hedgehog signaling remains to be defined in the future.

# Materials and Methods

## Cell lines, plasmids, and antibodies

hTERT-RPE1 cells were purchased from the ATCC, and 293T cells were a gift from Dr. Zhao Shan (Yunnan University). NME7 knockout cells were generated in the laboratory.

Mouse Nme7 and human NME7 coding regions were amplified from cDNA and ligated into a lentiviral pLVX backbone vector with a Flag tag at either the N or the C terminus. Nme7 mutants were generated by PCR amplification.

**Figure 4. NME7 knockout reduced primary cilium assembly.**
**(A)** Representative images of RPE1 WT and NME7 KO cells grown on glass coverslips after 48 h of serum starvation. NME7, red; Ac-Tub, green; DAPI, DNA in blue. White boxes highlight magnified regions. Scale bars, 10 and 2 μm in unmagnified and magnified images, respectively. **(B)** Immunofluorescence images of NME7 KO cells and control cells under normal growth conditions. Arrows indicate cells with primary cilium. ARL13b, red; Ac-Tub, green; DAPI, DNA in blue. Scale bar, 20 μm. **(C, D)** Quantification of cilium occurrence after NME7 knockout. Percentages of GT335[+] (C) and IFT88[+] (D) cilia are shown. **(E, F)** Time-course analysis of cilium occurrence after serum starvation for indicated hours after NME7 knockout. Percentages of ARL13b[+] (E) and Ac-Tub[+] (F) cilia are shown. **(G)** Percentages of ARL13b[+] cilia in control and NME7 knockout RPE1 cells without or with exogenous Nme7 expression.

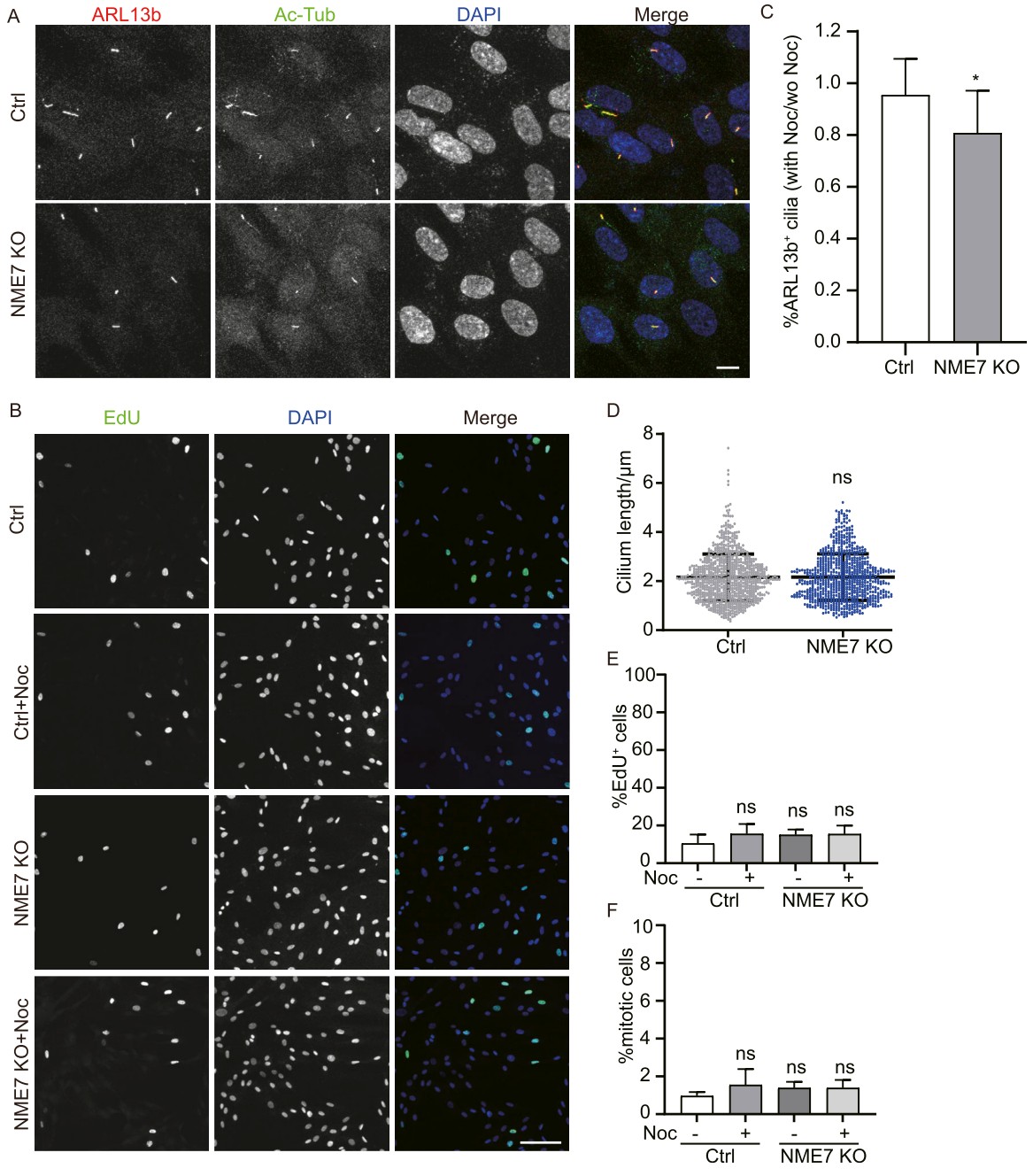

**Figure 5. Nocodazole treatment accelerated down-regulation of cilium formation in NME7 knockout cells.**
**(A)** Representative images of control and NME7 knockout 48-h serum-starved RPE1 cells with 1 µg/ml nocodazole treatment for 1 h. ARL13b, red; Ac-Tub, green; DAPI, DNA in blue. Scale bar, 10 µm. **(B)** EdU incorporation analysis of control and NME7 knockout RPE1 cells. Nocodazole treatments are indicated. EdU, green; DAPI, DNA in blue. Scale bar, 10 µm. **(C)** Quantification of the cilium occurrence under nocodazole compared to without nocodazole after NME7 knockout. Ratios of the percentages of ARL13b$^+$ cilia were quantified. **(D)** Quantification of cilium length in nocodazole-treated RPE1 cells after NME7 knockout. **(E, F)** Quantification of EdU$^+$ (E) and mitotic (F) cells in serum-starved RPE1 cells upon nocodazole treatment.

### Primary antibodies

The following primary antibodies were used: acetylated tubulin (66200-1-Ig; Proteintech), ARL13b (17711-1-AP; Proteintech), ARL13b (140076; Addgene) (Andrews et al, 2019), IFT88 (13967-1-AP; Proteintech), CP110 (12780-1-AP; Proteintech), NME7 (264 49-1-AP; Proteintech), Smo (sc-166685; Santa Cruz), β-actin (66009-1-Ig; Proteintech), α-tubulin (66031-1-Ig; Proteintech), INPP5E (17797-1-AP; Proteintech), Flag-M2 (F1804; Sigma-Aldrich), Flag (175359 and 175358; Addgene) (Peng et al, 2021), α-tubulin (ZMS1039; Sigma-Aldrich), GT335 (AG-20B-0020-C100; AdipoGen), γ-tubulin (T5326; Sigma-Aldrich), and CEP164 (22227-1-AP; Proteintech).

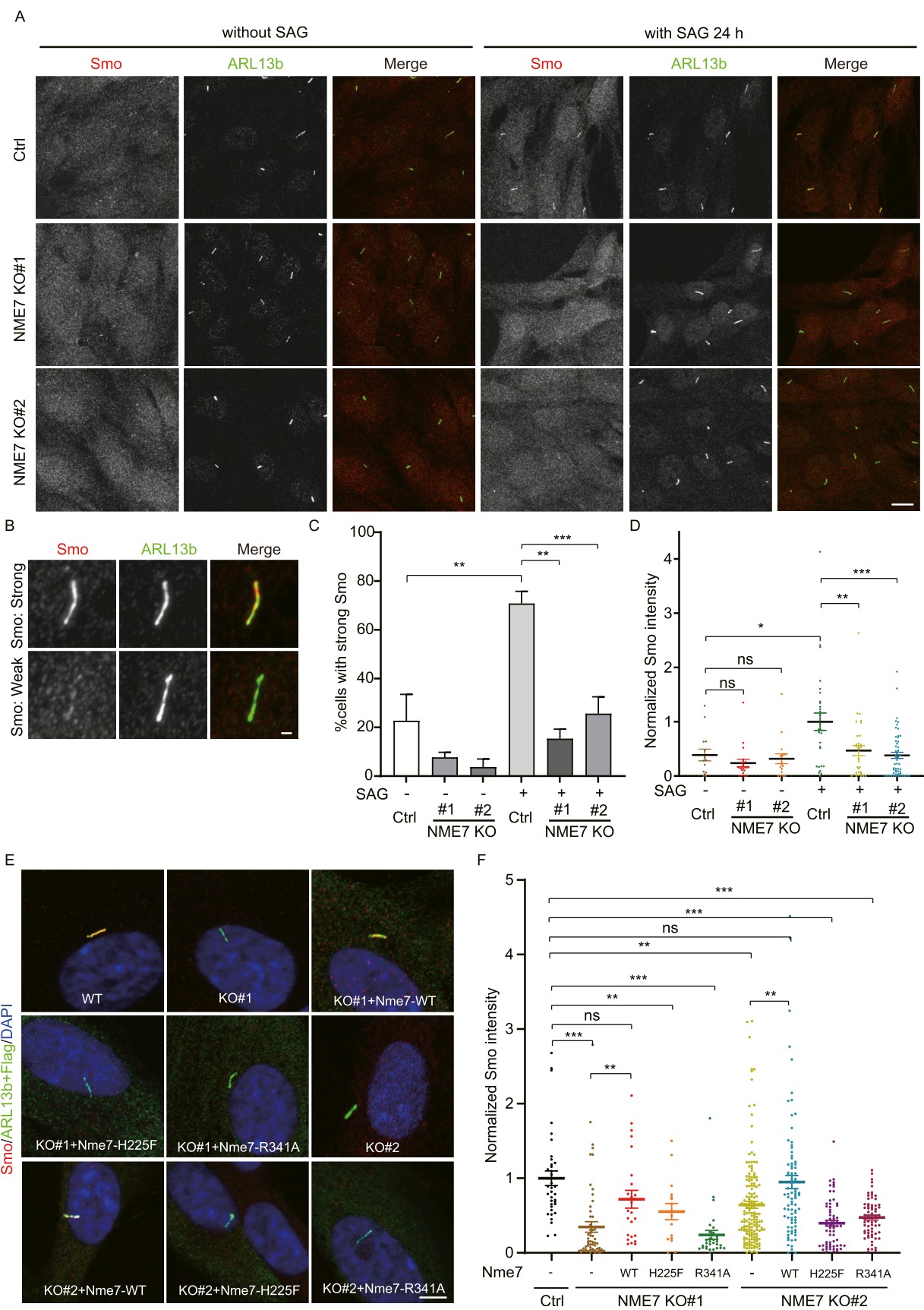

### Secondary antibodies

The Peroxidase AffiniPure Goat Anti-Rabbit/Mouse IgG (H+L) (111-035-003, 115-035-003), Alexa Fluor 488/594/647-AffiniPure Goat Anti-Mouse IgG (H+L) (115-545-146, 115-585-146, and 115-605-146), and Alexa Fluor 488/594/647 AffiniPure Goat Anti-Rabbit IgG (H+L) (111-545-144, 111-585-144, and 111-605-144) were from Jackson ImmunoResearch.

### Cell culture and cryopreservation

Cells were cultured in high-glucose DMEM supplemented with 10% FBS. The culture was maintained in a 5% CO2 atmosphere. For cell recovery from cryopreservation, vials were rapidly thawed in a 37°C water bath after removal from liquid nitrogen storage. Thawed cells were transferred to prewarmed culture medium in a dish and incubated under standard conditions. For cryopreservation, a specific medium containing DMEM, 20% FBS, and 10% DMSO was used.

### Lentivirus packaging and infection

For lentivirus packaging, we used 293T cells, which were cultured until reaching 70–80% confluency. The virus packaging mix, comprising plasmids (PSPAX2, VSVG) and PEI40K, was prepared and incubated at room temperature for 15 min. This mix was then added to the culture dish, ensuring thorough mixing with the culture medium. Subsequently, the cells were incubated at 37°C in a 5% CO2 incubator for 16 h. The first batch of virus was collected 48 h post-transfection and stored at 4°C, whereas the second batch was collected 72 h post-transfection using the same procedure and stored at –80°C.

For lentivirus infection, cells were plated 2 d before infection when they reached 20–30% confluency. The infection system consisted of DMEM, virus, and polybrene. The existing culture medium was replaced with the prepared infection medium, and cells were incubated at 37°C in a 5% CO2 incubator for 16 h.

### Immunofluorescence and microscopy

Cells were cultured on coverslips, the culture medium was aspirated, and 4% PFA was added to fix the cells. After 15 min of fixation, the PFA was aspirated and the cells underwent three washes with 1× DPBS. Next, permeabilization was achieved using 0.2% Triton X-100 at room temperature for 5 min, followed by an additional three washes with DPBS. For immunostaining, the primary antibody was diluted in a buffer containing PBS, 0.1% Triton X-100, and 3% BSA. The coverslips were incubated with the primary antibody for 2 h, followed by three washes with 1× DPBS. Subsequently, the secondary antibody was diluted in the same

buffer, and coverslips were incubated for an additional 2 h, followed by three washes with 1× DPBS. Finally, DAPI staining was performed by diluting DAPI in 1× DPBS. The stained cells were then mounted using Mowiol mounting medium. After air-drying in the dark, the slides were ready for fluorescence imaging and could be stored at 4°C.

Fluorescence imaging was performed using Zeiss LSM800 equipped with Airyscan and LSM980 equipped with Airyscan2 laser scanning confocal microscopes. Solid-state lasers at 405, 488, 561, and 639 nm were used to excite specific fluorophores. ZEN software (Zeiss) was used for image capture and processing. Maximum projection in the xy-plane was processed for multiple z-sectioned images.

### Western blotting

For sample preparation, cells were gently washed with precooled 1× DPBS. Subsequently, they were lysed using SDS sampling buffer. After a brief heating and sonication step, the lysates were stored at –20°C for further analysis. To analyze protein expression, an SDS–PAGE gel was prepared, with gel composition tailored to the molecular weight of the target protein. Protein samples were loaded onto the gel, and electrophoresis was conducted at a constant voltage of 80 V for 150 min. After electrophoresis, the proteins within the gel were transferred onto a PVDF membrane using a specific transfer buffer. This membrane served as a solid support for subsequent antibody-based detection. To minimize non-specific binding, the PVDF membrane was blocked using a solution of TTBS (Tris-buffered saline with Tween-20) containing 5% BSA. The membrane was then sequentially incubated with the primary antibody, followed by a secondary antibody. After each incubation step, thorough washes were performed to remove unbound antibodies. Finally, chemiluminescence imaging using enhanced chemiluminescence revealed the protein bands on the membrane.

### shRNA knockdown and CRISPR knockout

The sequences are as follows:

shNME7-F: CCGGGCTTCACTTCTTCGACGTTATCTCGAGATAACGTCGAAGAAGTGAAGCTTTTTG.

NME7 gRNA#1-F: CACCGACTGGTATTAATTGACTATG.

NME7 gRNA#2-F: CACCGACTTGAAAAAGGGTCTTGAC.

For gRNA plasmid construction, we selected target sequences using the CRISPOR website. The paired primers were annealed, and the vector was digested with BsmBI-V2. The annealed primers were ligated with the vector and transformed, and plasmids were extracted. To achieve CRISPR/Cas9 gene knockout, we packaged gRNA and Cas9 plasmids into lentiviruses. Low-passage RPE1 cells

---

**Figure 6. NME7 maintains proper Hh signaling in cilia.**
**(A)** Representative images of control and NME7 knockout RPE1 cells under 48 h of serum starvation without or with SAG treatment. The smoothened agonist SAG was added 24 h post-serum starvation and continued for another 24 h before cell fixation. Smo, red; ARL13b, green. Scale bar, 10 $\mu$m. **(B, C)** Percentages of ARL13b[+] cells with strong Smo fluorescence were quantified. Representative images of strong and weak Smo staining are indicated (B). Smo, red; ARL13b, green. Scale bar, 1 $\mu$m. **(D)** Normalized Smo intensity against cilium length in control and NME7 knockout RPE1 cells without or with SAG treatment as in (A). **(E)** Representative images of control and NME7 knockout RPE1 cells expressing indicated Nme7-Flag proteins under 48 h of serum starvation with 24-h SAG treatment. Smo, red; ARL13b, green. Scale bar, 5 $\mu$m. **(F)** Normalized Smo intensity against cilium length as in (E).

were infected with these lentiviruses, and single-clone selection was performed after 7 d. For this selection, cells were digested with trypsin, counted, and resuspended in culture medium. They were then distributed into a 96-well plate and cultured for 7 d. Clones were marked and allowed to continue growing. Once fully grown, the clones were digested, with some cryopreserved and others passaged for further detection. To identify edited clones, we digested them and centrifuged the cells. The supernatant was removed, and cells were resuspended in DNA extract buffer for PCR. The PCR products were run on a 2% agarose gel, and bands corresponding to edited sequences were sent for sequencing.

### EdU labeling

EdU labeling detection was performed using BeyoClick EdU-488 Cell Proliferation Detection Kit (C0071S; Beyotime) following the manufacturer's instructions. In brief, EdU was added to the culture medium to a final concentration of 10 $\mu$M. The cells were fixed with 4% PFA for 15 min and then washed three times with 1× DPBS. Cell membranes were permeabilized using 0.2% Triton X-100 for 10 min, followed by three washes with 1× DPBS. The click reaction solution was prepared according to the instructions and uniformly applied to the coverslips, incubating at room temperature and avoiding light for 30 min. After removing the reaction liquid, the cells were washed three times with 1x DPBS. Finally, cells were stained with DAPI for 5 min, followed by a DPBS wash. Coverslips were mounted onto slides using Mowiol for imaging. The proportion of EdU-positive cells relative to the total cell count was quantified to evaluate the proliferation rate.

### Quantification

Fluorescence image measurements were conducted using ZEN and ImageJ software. All statistical analyses were performed using GraphPad Prism software. Unless otherwise specified, a non-paired two-tailed $t$ test or one-way ANOVA was used for statistics, assuming normal Gaussian distribution for post hoc tests. Data were collected from at least three independent experiments in all cases.

## Data Availability

Requests for resources and reagents will be fulfilled by the corresponding author.

## Supplementary Information

## Acknowledgements

We thank the Imaging Facility of Center for Life Sciences, Yunnan University, for support. We thank Drs. Chuanmao Zhang (Peking University), Xueliang Zhu (Center for Excellence in Molecular Cell Science, Chinese Academy of Sciences), Zhao Shan (Yunnan University), Joost Snijder (Utrecht University), and James Trimmer (University of California, Davis) for sharing reagents. This work was supported by the National Natural Science Foundation of China (32070695) and Yunnan Province Basic Research Program (202301BF070001-012, 202101AW070016, 202201AT070071, and 202301AT070113). W Cui, Q Feng, H Zhu, and W Zhang were supported by the Graduate Student Research Program from Yunnan Provincial Department of Education and Yunnan University.

## Author Contributions

M Ji: conceptualization, data curation, validation, investigation, methodology, and writing—original draft, review, and editing.
W Cui: conceptualization, data curation, investigation, methodology, and writing—original draft, review, and editing.
Q Feng: conceptualization, data curation, validation, investigation, methodology, and writing—original draft, review, and editing.
J Qi: investigation and methodology.
X Wang: investigation and methodology.
H Zhu: investigation and methodology.
W Zhang: investigation and methodology.
W Fu: conceptualization, supervision, funding acquisition, investigation, and writing—original draft, review, and editing.

## Conflict of Interest Statement

The authors declare that they have no conflict of interest.

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
