## [Reviewer comments · Life Science Alliance]

Life Science Alliance

NME7 maintains primary cilium assembly, ciliary microtubule stability and Hedgehog signaling

Menghui Ji, Wenjuan Cui, Qian Feng, Jingjin Qi, Xinmin Wang, Hong Zhu, Wenqing Zhang, and Wenxiang Fu
DOI: <https://doi.org/10.26508/lsa.202402933>

Corresponding author(s): Wenxiang Fu, Yunnan University

Review Timeline:

Submission Date:	2024-07-08
Editorial Decision:	2024-08-08
Revision Received:	2024-12-11
Editorial Decision:	2025-01-02
Revision Received:	2025-01-05
Accepted:	2025-01-06

Transaction Report:

August 8, 2024

Re: Life Science Alliance manuscript #LSA-2024-02933-T

Dr. Wenxiang Fu
Yunnan University
China

Dear Dr. Fu,

Thank you for submitting your manuscript entitled "NME7 maintains proper ciliary microtubule stability and Hedgehog signaling" to Life Science Alliance. The manuscript was assessed by expert reviewers, whose comments are appended to this letter. We invite you to submit a revised manuscript addressing the Reviewer comments.

Thank you for this interesting contribution to Life Science Alliance. We are looking forward to receiving your revised manuscript.

Sincerely,

B. MANUSCRIPT ORGANIZATION AND FORMATTING:

Reviewer #1 (Comments to the Authors (Required)):

In this paper, Fu and colleagues dissect the function of the NME7 protein at basal bodies and cilia. They show that NME7 localizes to centrosomes and cilia, and knocking out NME7 modestly reduces ciliation. The study and results are clearly presented, although the conclusions would be strengthened by additional straightforward experimentation described below.

Points

1. It would be helpful if the authors could propose a function and mechanism of action for the NME7 protein. If NME7 is important for axonemal stability, they could, for example, test the localization of other tubulin binding proteins and tubulin post-translational modifications in KO cells. Also, what is the relationship between g-TURC association and the ciliary role of NME7 described here? Can the ciliation activity be rescued by constructs lacking the NDK/g-TURC interaction domains?
2. Where is NME7 localizing within the cilium? Is localization of other ciliary and Hh pathway proteins, besides SMO, affected in the knock-out?
3. Related to the above point, how does loss of NME7 affect ciliary structure? TEM or IF localization of other proteins at the basal body, distal appendage, and transition zone can be performed. These straightforward studies would add justification to the authors' conclusion that NME7 "plays a critical role in the formation and stability of primary cilia."

Reviewer #2 (Comments to the Authors (Required)):

NME7 maintains proper ciliary microtubule stability and Hedgehog signaling

Ji et al.

The manuscript by Ji et al. investigates the role of NME7 in cilia function, expanding on its previously identified roles as a component of the γ -TuRC, microtubule inner protein (MIP) (<https://doi.org/10.1016/j.cell.2023.05.009>), and in Wnt signaling (DOI: 10.1158/0008-5472.CAN-21-1020). In their study, Ji et al. primarily focus on the role of NME7 in cilia formation and signaling. They first confirm that NME7 localizes to centrosomes and the cytoplasm. By expressing mouse Nme7 in human cells, they validate its expression and centrosomal localization. In cells depleted of NME7 using siRNA, cilia formation following serum starvation did not indicate a significant role for NME7 in ciliation. However, due to only partial depletion, the authors generated NME7 knockout (KO) cell lines, where the NME7 fluorescence signal was completely absent. After 48 hours of serum starvation, these KO cells exhibited a moderate reduction in cilia formation compared to control cells, although no differences were observed at earlier time points.

Subsequently, the authors treated ciliated wild-type (WT) and NME7 KO cells with nocodazole to assess its impact on cilia stability. They found that nocodazole affected cilia stability more in NME7 KO cells, suggesting that axoneme microtubules are slightly less stable in the absence of NME7. Finally, the authors measured Hedgehog signaling, which relies on cilia formation, and noted that the two NME7 KO cell lines showed slight differences in this experiment. Focusing on cells with strong Smo fluorescence in the cilium, they observed differences between WT and NME7 KO cells.

The presented data are mostly not convincing as outlined below. The images are challenging to interpret, with cilia difficult to detect even on an enlarged screen (Figures 2B, 3A, and 4A), except for clear examples in Figure 4C and S2. The observed phenotypes are moderate, and there is a lack of molecular understanding of NME7's function in ciliation. There is also concern about the novelty of the results. Overall, the manuscript requires substantial revision to meet publication standards.

Major points

1. ", we found that Nme7/NME7 co-localizes with primary cilia in a small number of cells (Figure S2), suggesting". Figure S2 does not contain quantification just images of selected cells. What does small number of cells mean? Is this specific? What about non-overexpressed NME7?
2. Fig. 1G: no statistical difference - there is a high likelihood that the two samples do not show a difference. Thus, the slight decrease that is mentioned in the text is meaningless.
3. What is the purpose of the Nme7 overexpression experiment?
4. Quantification and statistics of Fig. S4 is missing.
5. Fig. 3B: since cilia formation is already reduced in NME7 KO cells, the authors should normalize cilia formation in WT and

NME7 KO cells to 1 and then measure the relative decrease in the presence of nocodazole. The present analysis is difficult to judge because of two variable parameters (NME7 KO and nocodazole).

6. The authors should perform a rescue experiment.

7. Fig. 4: Hedgehog was measured 72 h after serum starvation - Fig. 2 measured only 48 h. How does 72 h serum starvation affect cilia number and length in WT and NME7 KO cells?

Minor points

1. Fig. 1: the authors should show enlargements of the centrosomes in Fig. 1A.

2. Size of bands in Fig. S3G? Add marker.

Reviewer #3 (Comments to the Authors (Required)):

Ji, Fu and colleagues here examine the roles of the ciliary component, NME7 in ciliogenesis and cilium functioning. They describe a centrosomal/ basal body localization for the protein and then use reverse genetics to ask how its loss impacts on ciliogenesis. A moderate delay in primary ciliogenesis is described in the absence of NME7, along with some limitation of the localisation of Smoothed to cilia. The extent to which this study advances our understanding of NME7 is relatively limited. The data are not convincingly assembled; there are relatively few analyses presented and there are important controls missing from the experiments. Additional findings should be included for this work to make a useful contribution to the literature.

Major points

1. The localisation of NME7 (endogenous or overexpressed) to the centrosome/ basal body provides limited detail. More specific localisation experiments should be performed with additional co-markers for specific centriolar components and at different stages of the cell cycle outside mitosis.

2. It is unclear how relevant the ciliary localization of massively overexpressed Nme7/ NME7 is. What percentage of ciliated, Flag-positive cells show the (moderate) ciliary signal?

3. The ciliation defect described in Figs 2 and 3 is relatively moderate and an NME7 rescue experiment is needed to exclude off-target or other clonal effects. A similar control analysis is necessary for the experiment in Fig 4.

4. The data shown in Fig S4 are qualitative and are insufficient to support the conclusion that there is no difference in spindle MT stability. This experiment should be repeated with a quantitative analysis included.

5. It is possible that the alteration in cilium frequency seen in the nocodazole-treated cell population may be related to altered cell cycle distributions. An analysis of the cell cycle distribution in NME7 cells should be performed.

6. The analysis of the SMO localization to cilia in Fig 4 is not optimal. The calculation of SMO signal should be on a per-length basis for ARL13B, rather than per 'ARL-positive cell'. It is notably difficult to distinguish what is being quantified from the PDF version of the Figure that is available for review, so that Fig 4A should be markedly improved to demonstrate the basis for the assay.

7. Size markers must be included for immunoblots (Figs 1B, S1A, S1B, S3G).

Minor points

8. The data shown in Fig 2C-2F could be consolidated into a timecourse, which would indicate more clearly the kinetic nature of the putative ciliogenesis defect.

9. It is unclear what the boxed area in Fig 1A is for. The boxes do not help the reader. Similarly, the makeup of Fig 1B is unhelpful. What is shown here?

10. Figs 1C, 4C, S2, S3H, S4 should include less blank space bounding the cell/ ROI. The key information to be derived from the Figure should be made clearer.

11. Individual channels should be presented as monochrome images and only the merged images colored.

12. Fewer cells should be shown in Fig 1D; the arrows are not helpful in the Figure and could be deleted.

13. The PCRs shown in Fig S3A, S3B do not contribute usefully to the manuscript and should be deleted.

Referee Cross-Comments

There is good concurrence between the referees' reports. I consider that the other referees have provided reasonable and constructive comments.

(Note: re-reading the comments, I have changed the second 'qualitative' in my original point 4. to 'quantitative', which is the desired phrasing.)

Response Letter

We thank the editors and three reviewers for their efforts. We think that all the comments from the reviewers are constructive and helpful. We have taken these seriously and revised our manuscript accordingly. We believe that our new manuscript has been significantly strengthened and improved, and we hope this new version addresses the reviewers' concerns. Specific point-to-point responses are listed below.

Reviewer #1 (Comments to the Authors (Required)):

In this paper, Fu and colleagues dissect the function of the NME7 protein at basal bodies and cilia. They show that NME7 localizes to centrosomes and cilia, and knocking out NME7 modestly reduces ciliation. The study and results are clearly presented, although the conclusions would be strengthened by additional straightforward experimentation described below.

Response:

We thank you for your constructive comments. In our revised manuscript (now with new main Figures 1-6 and new supplemental Figures S1-S5), we have made changes mainly in three aspects:

- 1) We carefully examined the localization of endogenous and exogenous NME7, involving co-staining with various cilium/centrosome markers, and concluded that NME7 localizes at centrosomes, basal bodies, and cilia.
- 2) The conclusions regarding NME7 in cilium assembly and Hedgehog signaling were strengthened by additional experiments, and we further suggest that the role of NME7 in Hedgehog signaling may rely on NDPK activity and γ -TuRC association.
- 3) The data in the revised manuscript are now more clearly presented.

Points

1. It would be helpful if the authors could propose a function and mechanism of action for the NME7 protein. If NME7 is important for axonemal stability, they could, for example, test the localization of other tubulin binding proteins and tubulin post-translational modifications in KO cells. Also, what is the relationship between g-TURC association and the ciliary role of NME7 described here? Can the ciliation activity be rescued by constructs lacking the NDK/g-TURC interaction domains?

Response:

1) While some antibodies were not working, we successfully examined ARL13b (ciliary membrane protein), Ac-tub (acetylated tubulin, axonemal microtubule marker), GT335 (polyglutamylated tubulin), IFT88 (transition zone and axoneme), and CEP164 (distal appendage) in NME7 KO cells. The morphologies of these markers were not affected (new Figures 4A, S4C-4E). Time-course analysis of primary cilium assembly post serum starvation indicated reductions in cilium assembly 48 hours post serum starvation but not at earlier time points (new Figures 4C-4F). We suggest that the function of NME7 is not required for core structures (such as the distal appendage and transition zone) for cilium initiation.

2) In an effort to pursue the mechanism of NME7 action, and since the Hedgehog defects were much stronger and clearer than the relatively modest cilia assembly defects, we performed the Hedgehog signaling rescue experiments using NDPK/ γ -TuRC interaction mutants. Data showed (new Figures 6E and 6F) that only the wild type, and not the mutants, could rescue the Hedgehog signaling defects caused by NME7 knockout. Together, these results indicate that NME7 is important for ciliary Hedgehog signaling, likely requiring its NDPK activity and γ -TuRC formation. While GTP is well-known to be required for microtubule assembly and γ -TuRC forms the microtubule organizing center, it is reasonable to suggest that NME7 may locally produce GTP via γ -TuRC complex formation, thus promoting ciliary microtubule stability and Hedgehog signaling. These points have also been discussed in the manuscript.

2. Where is NME7 localizing within the cilium? Is localization of other ciliary and Hh pathway proteins, besides SMO, affected in the knock-out?

Response:

1) We re-examined our previous data and performed additional high-resolution microscopy imaging experiments using Airyscan2 to check the localization of NME7 within cilia. We further confirmed that both endogenous and exogenous NME7 localize to basal bodies and cilia. While the NME7 staining pattern in most cases is shorter than the entire cilium, we suggest that NME7 likely resides along the axonemal microtubule. See the below texts in the manuscript for details.

“While we noticed evident NME7 staining decorating spindle microtubules besides the spindle poles during mitosis (Fig 1A), we next asked if NME7 is also

present within cilium. Using high-resolution microscopy, we detected the staining of NME7, but not CEP164 (centrosome protein 164, a typical distal appendage marker), within primary cilium (Fig 2A, also shown in Fig 1C and 1D). The endogenous NME7 tends to have stronger staining signals at the proximal part of the primary cilium. We then tested the localization of overexpressed mouse Nme7 and human NME7, and also confirmed the co-localization of Flag tagged Nme/NME7 with primary cilium markers (Fig 2B-E). While the NME7 staining pattern in most cases is shorter than the entire cilium, we suggest that NME7 likely resides along the axonemal microtubule. It is worth mentioning that NME7 signals at the centrosomes is much stronger than with the cilium, and we only observed ~10% cells showing indicated ciliary staining. However, due to limitation in microscopy resolution, and high background caused by endogenous antibody quality or overexpression, we suggest that the percentages of cells with ciliary localization of NME7 could be much underestimated. For example, staining of exogenously expressed NME7 with the cilium can only be observed when the cilium was clearly viewed as emanating from the cell surface due to high intracellular background.”

2) This question has also been raised in point 1. See the response to point 1. ARL13b, Ac-tub, GT335, IFT88, CEP164, and Smo were tested. We suggest that the ciliary structure (including the distal appendage and transition zone) was not affected. Rather, the percentages of cilia represented by various markers and Hedgehog signaling were decreased in NME7 KOs.

3. Related to the above point, how does loss of NME7 affect ciliary structure? TEM or IF localization of other proteins at the basal body, distal appendage, and transition zone can be performed. These straightforward studies would add justification to the authors' conclusion that NME7 "plays a critical role in the formation and stability of primary cilia."

Response:

Also see the response to point 1. NME7 does not affect the basic structure of cilia as shown by immunofluorescence. Basal body (γ -Tubulin), distal appendage (CEP164), and transition zone (IFT88, which also marks the axoneme) were demonstrated by IF. However, we have difficulties in TEM and immuno-EM sample preparation that are hard to overcome in a short time.

The cilia in NME7 KO cells were more sensitive to nocodazole treatment

compared with the control (Figures 5A, 5C, and 5D). This conclusion was also strengthened by analysis of the cell cycle (Figures 5B, 5E, and 5F) and spindle microtubule stability (Figures S5A and S5B).

Reviewer #2 (Comments to the Authors (Required)):

The manuscript by Ji et al. investigates the role of NME7 in cilia function, expanding on its previously identified roles as a component of the γ -TuRC, microtubule inner protein (MIP) (<https://doi.org/10.1016/j.cell.2023.05.009>), and in Wnt signaling (DOI: 10.1158/0008-5472.CAN-21-1020). In their study, Ji et al. primarily focus on the role of NME7 in cilia formation and signaling. They first confirm that NME7 localizes to centrosomes and the cytoplasm. By expressing mouse Nme7 in human cells, they validate its expression and centrosomal localization. In cells depleted of NME7 using siRNA, cilia formation following serum starvation did not indicate a significant role for NME7 in ciliation. However, due to only partial depletion, the authors generated NME7 knockout (KO) cell lines, where the NME7 fluorescence signal was completely absent. After 48 hours of serum starvation, these KO cells exhibited a moderate reduction in cilia formation compared to control cells, although no differences were observed at earlier time points.

Subsequently, the authors treated ciliated wild-type (WT) and NME7 KO cells with nocodazole to assess its impact on cilia stability. They found that nocodazole affected cilia stability more in NME7 KO cells, suggesting that axoneme microtubules are slightly less stable in the absence of NME7. Finally, the authors measured Hedgehog signaling, which relies on cilia formation, and noted that the two NME7 KO cell lines showed slight differences in this experiment. Focusing on cells with strong Smo fluorescence in the cilium, they observed differences between WT and NME7 KO cells.

Response:

We extend our gratitude for your detailed review and constructive comments.

The presented data are mostly not convincing as outlined below. The images are challenging to interpret, with cilia difficult to detect even on an enlarged screen (Figures 2B, 3A, and 4A), except for clear examples in Figure 4C and S2. The observed phenotypes are moderate, and there is a lack of molecular understanding of NME7's function in ciliation. There is also concern about the novelty of the results.

Overall, the manuscript requires substantial revision to meet publication standards.

Response:

1) We apologize for the low image quality presented in our initially submitted manuscript, which was further dampened by PDF compression in the submission system. We have now re-prepared the figures and believe this revised manuscript is much more clearly presented. For example, your mentioned old Figures 2B, 3A, and 4A are now new Figures 4A, 5A, and 6A, respectively.

2) We agree that the role of NME7 in cilium assembly is moderate. We suggest that NME7 maintains cilia at the mature stage and cilium stability. We performed additional experiments to enhance the novelty of this study and gain a deeper understanding of NME7's action (now with new main Figures 1-6 and new supplemental Figures S1-S5).

We stained additional markers in NME7 knockouts to examine how the cilia are affected. In our efforts to obtain more details regarding the localization of NME7, we carefully examined the localization of endogenous and exogenous NME7, involving co-staining with various cilium/centrosome markers, and concluded that NME7 localizes at centrosomes, basal bodies, and cilia. Additionally, while the Hedgehog phenotype is much stronger, we performed rescue experiments using NME7 mutants and suggest that NME7's function may rely on its NDPK activity and γ -TuRC association (Also see responses to your specific points).

Major points

1. ", we found that Nme7/NME7 co-localizes with primary cilia in a small number of cells (Figure S2), suggesting". Figure S2 does not contain quantification just images of selected cells. What does small number of cells mean? Is this specific? What about non-overexpressed NME7?

Response:

It is now new Figure 2. The localization of endogenous and exogenous NME7 was confirmed within cilia. The observed localization rate is approximately 10%, but we believe this number is significantly underestimated. Bleed-through from other channels can be ruled out since the staining pattern of NME7 along cilia in nearly all cases does not 100% match the ciliary marker. CEP164, used as a negative control, did not show any ciliary signals. See the below texts in the manuscript for details.

“While we noticed evident NME7 staining decorating spindle microtubules

besides the spindle poles during mitosis (Fig 1A), we next asked if NME7 is also present within cilium. Using high-resolution microscopy, we detected the staining of NME7, but not CEP164 (centrosome protein 164, a typical distal appendage marker), within primary cilium (Fig 2A, also shown in Fig 1C and 1D). The endogenous NME7 tends to have stronger staining signals at the proximal part of the primary cilium. We then tested the localization of overexpressed mouse Nme7 and human NME7, and also confirmed the co-localization of Flag tagged Nme/NME7 with primary cilium markers (Fig 2B-E). While the NME7 staining pattern in most cases is shorter than the entire cilium, we suggest that NME7 likely resides along the axonemal microtubule. It is worth mentioning that NME7 signals at the centrosomes is much stronger than with the cilium, and we only observed ~10% cells showing indicated ciliary staining. However, due to limitation in microscopy resolution, and high background caused by endogenous antibody quality or overexpression, we suggest that the percentages of cells with ciliary localization of NME7 could be much underestimated. For example, staining of exogenously expressed NME7 with the cilium can only be observed when the cilium was clearly viewed as emanating from the cell surface due to high intracellular background.”

2. Fig. 1G: no statistical difference - there is a high likelihood that the two samples do not show a difference. Thus, the slight decrease that is mentioned in the text is meaningless.

Response:

We agree with your points. The text has been revised to “...revealed that NME7 knockdown did not cause significant change in the primary cilium assembly (Fig 3B-3E).”

3. What is the purpose of the Nme7 overexpression experiment?

Response:

In our revised manuscript, the localization of overexpressed Nme7 matches that of endogenous NME7. We further performed rescue experiments using overexpressed Nme7.

4. Quantification and statistics of Fig. S4 is missing.

Response:

We quantified four different concentrations (0, 10 ng/ml, 50 ng/ml, and 500 ng/ml) of nocodazole in our revised manuscript (new Figure S5). See the figure and the corresponding text in the manuscript for details.

“To verify the effects of NME7 on stability of mitotic spindle microtubules, we next tested the sensitivity changes of spindle microtubules in NME7 KO to the microtubule depolymerizing drug nocodazole. Therefore, RPE1 cells and NME7 KO cells in the presence of different concentrations of nocodazole for 2 hours were analyzed (Fig S5A and S5B). The spindle microtubules underwent different degrees of depolymerization with the treatment of 0-500 ng/ml nocodazole, however the sensitivity of spindle microtubules to nocodazole in NME7 KO cells was similar to that in WT cells.”

5. Fig. 3B: since cilia formation is already reduced in NME7 KO cells, the authors should normalize cilia formation in WT and NME7 KO cells to 1 and then measure the relative decrease in the presence of nocodazole. The present analysis is difficult to judge because of two variable parameters (NME7 KO and nocodazole).

Response:

We agree. The data were normalized against the condition without nocodazole as a ratio to judge only the effects of nocodazole (new Figure 5). The potential effects of nocodazole on cell cycle changes in our experimental settings were further excluded.

6. The authors should perform a rescue experiment.

Response:

We have now performed cilia assembly and Hedgehog signaling rescue experiments (new Figures 6E and 6F). While the effects of NME7 on cilia percentage are modest compared with Hedgehog signaling defects, we further tested Hedgehog signaling using NME7 mutants to provide more molecular details regarding NME7.

7. Fig. 4: Hedgehog was measured 72 h after serum starvation - Fig. 2 measured only 48 h. How does 72 h serum starvation affect cilia number and length in WT and NME7 KO cells?

Response:

We apologize for the misunderstanding and unclear description in our previous manuscript. The assay point for Hedgehog was 48 hours post serum starvation, which

is also the most widely used condition in the literature. The text and figure legends have been made clear about this.

Text: “Both wild-type and NME7 KO RPE1 cells were subjected to serum starvation for 24 hours, and the cells were further kept in the serum starved medium without or with the Smo (smoothened) agonist SAG for an additional 24 hours.”

Figure legends: “The smoothened agonist SAG was added 24 hours post serum starvation and continued for another 24 hours before cell fixation.”

Minor points

1. *Fig. 1: the authors should show enlargements of the centrosomes in Fig. 1A.*

Response:

The enlargements are shown in new Figure 1A and nearly all the other figures for better presentation of our data.

2. *Size of bands in Fig. S3G? Add marker.*

Response:

The molecular markers are shown in new Figures S1A, S1B, and S2A.

Reviewer #3 (Comments to the Authors (Required)):

Ji, Fu and colleagues here examine the roles of the ciliary component, NME7 in ciliogenesis and cilium functioning. They describe a centrosomal/ basal body localization for the protein and then use reverse genetics to ask how its loss impacts on ciliogenesis. A moderate delay in primary ciliogenesis is described in the absence of NME7, along with some limitation of the localisation of Smoothened to cilia.

Response:

We extend our gratitude for your detailed review and constructive comments.

The extent to which this study advances our understanding of NME7 is relatively limited. The data are not convincingly assembled; there are relatively few analyses presented and there are important controls missing from the experiments. Additional findings should be included for this work to make a useful contribution to the literature.

Response:

The data are more clearly assembled in our revised manuscript, and rescue experiments were performed to obtain more convincing results. We conducted additional experiments to advance our understanding of NME7 (now with new main Figures 1-6 and new supplemental Figures S1-S5). We stained additional markers in NME7 knockouts to examine how the cilia are affected. In our efforts to get more details regarding the localization of NME7, we carefully examined the localization of endogenous and exogenous NME7, involving co-staining with various cilium/centrosome markers, and concluded that NME7 localizes at centrosomes, basal bodies, and cilia. In addition, while the Hedgehog phenotype is much stronger, we performed rescue experiments using NME7 mutants and suggest that NME7's function may rely on its NDPK activity and γ -TuRC association.

Major points

1. The localisation of NME7 (endogenous or overexpressed) to the centrosome/ basal body provides limited detail. More specific localisation experiments should be performed with additional co-markers for specific centriolar components and at different stages of the cell cycle outside mitosis.

Response:

We co-stained NME7 with additional centrosome/basal body markers in both growing and serum-starved cells (new Figures 1B-1D). Data showed that NME7 consistently co-localizes with centrosomal protein γ -tubulin under different centrosome duplication and separation stages. The majority of NME7 is localized at the basal body during the ciliation stage. See the below texts in the manuscript for details.

“In addition to showing a diffuse distribution in the cytoplasm, NME7 has a strong staining at the centrosome. We further examined the colocalization of NME7 co-stained with γ -tubulin in both interphase and mitosis (Fig 1B). As shown, NME7 consistently co-localizes with centrosomal protein γ -tubulin under different centrosome duplication and separation stages. With serum deprivation, RPE1 exits from cell cycle and assembles primary cilia. ARL13b (ADP-ribosylation factor-like protein 13b) labels the ciliary membrane and GT335 (polyglutamylated tubulin) indicates the ciliary axoneme. We examined NME7 with indicated centrosomal and ciliary markers in both non-ciliated and ciliated cells, and confirmed that NME7 is localized at the basal body (Fig 1C and 1D). Together, these support that NME7

localizes to the centrosome and basal body.”

2. It is unclear how relevant the ciliary localization of massively overexpressed Nme7/NME7 is. What percentage of ciliated, Flag-positive cells show the (moderate) ciliary signal?

Response:

It is now new Figure 2. The localization of both endogenous and exogenous NME7 was confirmed within cilia. The observed localization rate is approximately 10%, but we believe this number is significantly underestimated. Bleed-through from other channels can be ruled out since the staining pattern of NME7 along cilia in nearly all cases does not 100% match the ciliary marker. CEP164, used as a negative control, did not show any ciliary signals. See the below texts in the manuscript for details.

“While we noticed evident NME7 staining decorating spindle microtubules besides the spindle poles during mitosis (Fig 1A), we next asked if NME7 is also present within cilium. Using high-resolution microscopy, we detected the staining of NME7, but not CEP164 (centrosome protein 164, a typical distal appendage marker), within primary cilium (Fig 2A, also shown in Fig 1C and 1D). The endogenous NME7 tends to have stronger staining signals at the proximal part of the primary cilium. We then tested the localization of overexpressed mouse Nme7 and human NME7, and also confirmed the co-localization of Flag tagged Nme/NME7 with primary cilium markers (Fig 2B-E). While the NME7 staining pattern in most cases is shorter than the entire cilium, we suggest that NME7 likely resides along the axonemal microtubule. It is worth mentioning that NME7 signals at the centrosomes is much stronger than with the cilium, and we only observed ~10% cells showing indicated ciliary staining. However, due to limitation in microscopy resolution, and high background caused by endogenous antibody quality or overexpression, we suggest that the percentages of cells with ciliary localization of NME7 could be much underestimated. For example, staining of exogenously expressed NME7 with the cilium can only be observed when the cilium was clearly viewed as emanating from the cell surface due to high intracellular background.”

3. The ciliation defect described in Figs 2 and 3 is relatively moderate and an NME7 rescue experiment is needed to exclude off-target or other clonal effects. A similar

control analysis is necessary for the experiment in Fig 4.

Response:

We have obtained two different NME7 KO cells with different genotypes showing similar defects. We also performed rescue experiments using mouse Nme7 to exclude off-target or clonal effects. While the Hedgehog signaling defects were much stronger, we performed rescue experiments using both the wild type and mutants as well.

4. The data shown in Fig S4 are qualitative and are insufficient to support the conclusion that there is no difference in spindle MT stability. This experiment should be repeated with a quantitative analysis included.

Response:

We quantified four different concentrations (0, 10 ng/ml, 50 ng/ml, and 500 ng/ml) of nocodazole in our revised manuscript (new Figure S5). See the figure and the corresponding text in the manuscript for details.

“To verify the effects of NME7 on stability of mitotic spindle microtubules, we next tested the sensitivity changes of spindle microtubules in NME7 KO to the microtubule depolymerizing drug nocodazole. Therefore, RPE1 cells and NME7 KO cells in the presence of different concentrations of nocodazole for 2 hours were analyzed (Fig S5A and S5B). The spindle microtubules underwent different degrees of depolymerization with the treatment of 0-500 ng/ml nocodazole, however the sensitivity of spindle microtubules to nocodazole in NME7 KO cells was similar to that in WT cells.”

5. It is possible that the alteration in cilium frequency seen in the nocodazole-treated cell population may be related to altered cell cycle distributions. An analysis of the cell cycle distribution in NME7 cells should be performed.

Response:

In our analysis, the cells were serum starved for 48 hours to induce quiescence, and nocodazole was used for only 1 hour. We checked both the S-phase and mitotic phases, and neither was affected, which could contribute to the differences in cilium frequency. See the below texts in the manuscript for details.

“Percentages of S-phase (indicated by EdU incorporation) and mitosis were not affected by NME7 knockout or nocodazole treatment in the cilium assembly assay

conditions (Fig 5E and 5F). As a note, since the cells were serum starved for a long time (48 h) to induce quiescence and only treated with nocodazole for a short period (1 h), the percentages of mitotic cells in all cases were low (<2%). Together these results suggest that NME7 may maintain ciliary microtubule stability and this is unrelated to the cell cycle changes.”

6. The analysis of the SMO localization to cilia in Fig 4 is not optimal. The calculation of SMO signal should be on a per-length basis for ARL13B, rather than per 'ARL-positive cell'. It is notably difficult to distinguish what is being quantified from the PDF version of the Figure that is available for review, so that Fig 4A should be markedly improved to demonstrate the basis for the assay.

Response:

The Smo intensities are now normalized against the length of cilia in the revised manuscript (new Figures 6D and 6F). The original field was too large, and due to PDF compression in the submission system, we present the mentioned Figure 4A as new Figure 6A more clearly. Other figures are also better presented in the revised manuscript.

7. Size markers must be included for immunoblots (Figs 1B, S1A, S1B, S3G).

Response:

Size markers are included in the revised version, now in new Figures S1A, S1B, S2A, and S4A.

Minor points

8. The data shown in Fig 2C-2F could be consolidated into a timecourse, which would indicate more clearly the kinetic nature of the putative ciliogenesis defect.

Response:

A time course presentation of cilia occurrence is now included in the revised manuscript (new Figure 4E).

9. It is unclear what the boxed area in Fig 1A is for. The boxes do not help the reader. Similarly, the makeup of Fig 1B is unhelpful. What is shown here?

Response:

The presentation of all the figures, except for those you mentioned, is improved.

The old Figure 1A is now new Figure 1A. The makeup of Figure 1D (mentioned as 1B in comments) was removed and is now presented as new Figure 3A.

10. Figs 1C, 4C, S2, S3H, S4 should include less blank space bounding the cell/ ROI. The key information to be derived from the Figure should be made clearer.

Response:

The presentation of all the figures has been improved:
old Fig 1C is now new Figure S2B;
old Fig 4C is now new Figure 6B
old Fig S2 is now new Figures 2B-2E
old Fig S3H is now new Figure S4B
old Fig S4 is now new Figure S5A (since we re-did the quantification, only four concentrations of nocodazole were involved in the new manuscript.)

11. Individual channels should be presented as monochrome images and only the merged images colorized.

Response:

In all cases, the individual channels are now presented as monochrome images, making the data presentation clearer.

12. Fewer cells should be shown in Fig 1D; the arrows are not helpful in the Figure and could be deleted.

Response:

The old figure was enlarged and cropped, and the arrows were deleted. It is now presented as new Figure 3A for better clarity.

13. The PCRs shown in Fig S3A, S3B do not contribute usefully to the manuscript and should be deleted.

Response:

We agree, and they were deleted. The characterization of NME7 KO sequences is now presented as new Figure S3.

Referee Cross-Comments

There is good concurrence between the referees' reports. I consider that the other

referees have provided reasonable and constructive comments.

(Note: re-reading the comments, I have changed the second 'qualitative' in my original point 4. to 'quantitative', which is the desired phrasing.)

Response:

We thank the three anonymous reviewers for their constructive comments, which are in overall concurrence and have significantly helped improve our manuscript.

January 2, 2025

RE: Life Science Alliance Manuscript #LSA-2024-02933-TR

Dr. Wenxiang Fu
Yunnan University
School of Life Sciences
Kunming 650500
Barbados

Dear Dr. Fu,

Thank you for submitting your revised manuscript entitled "NME7 maintains primary cilium assembly, ciliary microtubule stability and Hedgehog signaling". We would be happy to publish your paper in Life Science Alliance pending final revisions necessary to meet our formatting guidelines.

- please address the remaining Reviewer comments
- please be sure that the authorship listing and order is correct
- please add the Twitter handle of your host institute/organization as well as your own or/and one of the authors in our system
- please rename the Resource Availability section as Data Availability

A. FINAL FILES:

B. MANUSCRIPT ORGANIZATION AND FORMATTING:

Sincerely,

Reviewer #1 (Comments to the Authors (Required)):

The authors have addressed all of my comments.

Reviewer #2 (Comments to the Authors (Required)):

The manuscript has been significantly improved, and many of my concerns have been addressed. However, I have two additional points that the authors should address before publication:

1. The Y-axis in Supplementary Fig. 5B is not labeled.
2. In Fig. 6F, the two NME7 KO cell lines exhibit differing behaviors in response to the NME7 mutants. Specifically, the H225F mutant enhances ciliation in KO clone 1 but reduces ciliation in KO clone 2. The authors should provide commentary on this discrepancy. Additionally, the meaning of the statistics presented in this subfigure is unclear. What is the reference point for comparison - the KO cell line? This should be explicitly indicated either in the figure or the legend.

Reviewer #3 (Comments to the Authors (Required)):

Ji, Cui, Feng, Fu and colleagues here present an analysis of NME7, a member of the nucleoside diphosphate kinase (NDPK) family, in primary cilium formation and functioning.

They demonstrate NME7 localisation to centrosomes and basal body and describe overexpressed, tagged NME7 as having a similar localisation. A localisation to cilia was observed for a fraction of endogenous NME7, with the overexpressed protein also being found there. No distinct phenotype having been observed in NME7 knockdown cells, NME7 CRISPR knockout cells were used to explore the impact of NME7 loss on ciliogenesis frequencies. A moderate decline was observed, from c. 80% to 55% at a fixed timepoint, which was rescued by overexpression of the murine Nme7 isoform, and a time-course experiment indicates a small difference in ciliation at 48h post-serum starvation. A moderate reduction was seen in the level of ciliation in NME7 null cells after nocodazole treatment. There was a convincing effect on Hh signalling in NME7-deficient cells that was rescued by wild-type, but not kinase-deficient NME7

The paper is notably improved on the initial submission and the findings are coherently presented. However, the conclusions drawn are overstated. From the data presented here, there does not appear to be a 'a pivotal role in the assembly and function of primary cilia' for NME7; nor is it convincing that NME7 'is essential for this [cilium assembly] process'. The proposed axonemal microtubule localisation is not convincingly demonstrated; nor is the decoration of spindle microtubules by NME7. The proposed localisation of NME7 to spindle microtubules is difficult to discern in Fig. 1A. The authors properly qualify their observations to note the high background staining that may obscure their conclusions, but this background also impedes their ability to derive a mechanism for the problems in Hh signalling seen in the absence of NME7. Much of the discussion is somewhat speculative,

given the limited impact seen in the experiments described here. The authors should revise how they describe/ discuss their data to reduce the overinterpretation of the relatively mild phenotypes seen here.

Other points:

1. The extent to which NME7 is knocked down should be quantitated from Fig. S2.
2. Figure 4E should present the x-axis as a timecourse, rather than as categories. This is the principal demonstration of any impact of NME7 deficiency on ciliary stability and it needs to be presented clearly.

2nd Authors' response***Reviewers' comments***

Reviewer #1 (Comments to the Authors (Required)):

The authors have addressed all of my comments.

Response:

We greatly appreciate your valuable and insightful feedback, which has substantially enhanced the quality of our manuscript.

Reviewer #2 (Comments to the Authors (Required)):

The manuscript has been significantly improved, and many of my concerns have been addressed.

However, I have two additional points that the authors should address before publication:

Response:

We are thankful for your detailed and valuable feedback, which has led to substantial improvements in our manuscript. These two additional points have been addressed.

1. The Y-axis in Supplementary Fig. 5B is not labeled.

Response:

We apologize for missing the label. The label "Relative microtubule intensity" was added to this Supplemental Figure.

2. In Fig. 6F, the two NME7 KO cell lines exhibit differing behaviors in response to the NME7 mutants. Specifically, the H225F mutant enhances ciliation in KO clone 1 but reduces ciliation in KO clone 2. The authors should provide commentary on this discrepancy. Additionally, the

meaning of the statistics presented in this subfigure is unclear. What is the reference point for comparison - the KO cell line? This should be explicitly indicated either in the figure or the legend.

Response:

Thank you for pointing this out. We have now indicated and clarified the statistics for the comparison in the figure.

As you noticed, H225F-KO1 values appeared higher than the KO1 group. This might be due to the relatively low number of cells quantified in the H225F-KO1 group, which could be a result of lower transfection efficiency in this paired experiment. Nonetheless, as shown in the figure, the two wild-type rescue groups are not significantly different from the control group. However, the rescue groups for both KO clones, whether with H225F or R341A mutant, are significantly lower than the control group. We have also added this commentary in our results section.

Reviewer #3 (Comments to the Authors (Required)):

Ji, Cui, Feng, Fu and colleagues here present an analysis of NME7, a member of the nucleoside diphosphate kinase (NDPK) family, in primary cilium formation and functioning.

They demonstrate NME7 localisation to centrosomes and basal body and describe overexpressed, tagged NME7 as having a similar localisation. A localisation to cilia was observed for a fraction of endogenous NME7, with the overexpressed protein also being found there. No distinct phenotype having been observed in NME7 knockdown cells, NME7 CRISPR knockout cells were used to explore the impact of NME7 loss on ciliogenesis frequencies. A moderate decline was observed, from c. 80% to 55% at a fixed timepoint, which was rescued by overexpression of the murine Nme7 isoform, and a time-course experiment indicates a small difference in ciliation at 48h post-serum starvation. A moderate reduction was seen in the level of ciliation in NME7 null cells after nocodazole treatment. There was a convincing effect on Hh signalling in NME7-deficient cells that was rescued by wild-type, but not kinase-deficient NME7

The paper is notably improved on the initial submission and the findings are coherently presented. However, the conclusions drawn are overstated. From the data presented here, there does not appear to be a 'a pivotal role in the assembly and function of primary cilia' for NME7; nor is it convincing that NME7 'is essential for this [cilium assembly] process'. The proposed axonemal microtubule localisation is not convincingly demonstrated; nor is the decoration of spindle microtubules by NME7. The proposed localisation of NME7 to spindle microtubules is difficult to discern in Fig. 1A. The authors properly qualify their observations to note the high background staining that may obscure their conclusions, but this background also impedes their ability to derive a mechanism for the problems in Hh signalling seen in the absence of NME7. Much of the discussion is somewhat speculative, given the limited impact seen in the experiments described here. The authors should revise how they describe/ discuss their data to reduce the overinterpretation of the relatively mild phenotypes seen here.

Reply:

We deeply appreciate your detailed and valuable feedback, which has greatly improved our manuscript. We have meticulously reviewed our text and revised our descriptions and discussions to avoid overinterpretation. We have minimized or revised speculative comments to present our data more objectively. For example, “we noticed evident NME7 staining decorating spindle microtubules besides the spindle poles” has been changed into “we noticed faint NME7 staining

along the spindle microtubules besides the spindle poles”. Terms like “pivotal”, “essential” and “crucial” were deleted. “NME7 likely resides along the axonemal microtubules” in the results section was removed. We also minimized speculative comments to present our data more objectively.

Other points:

1. The extent to which NME7 is knocked down should be quantitated from Fig. S2.

Reply:

The knockdown efficiency from various experiments was quantified and described in the text. As shown in the new Figure S2B, the NME7 levels in knockdown cells were approximately 26.0% of those in control cells.

2. Figure 4E should present the x-axis as a timecourse, rather than as categories. This is the principal demonstration of any impact of NME7 deficiency on ciliary stability and it needs to be presented clearly.

Reply:

Thank you for pointing this out. Both Figures 4E and 4F have been reorganized into a timecourse presentation.

January 6, 2025

RE: Life Science Alliance Manuscript #LSA-2024-02933-TRR

Dr. Wenxiang Fu
Yunnan University
School of Life Sciences
Kunming 650500
China

Dear Dr. Fu,

Thank you for submitting your Research Article entitled "NME7 maintains primary cilium assembly, ciliary microtubule stability and Hedgehog signaling". It is a pleasure to let you know that your manuscript is now accepted for publication in Life Science Alliance. Congratulations on this interesting work.

DISTRIBUTION OF MATERIALS:

Again, congratulations on a very nice paper. I hope you found the review process to be constructive and are pleased with how the manuscript was handled editorially. We look forward to future exciting submissions from your lab.

Sincerely,
